# Healthcare Contacts Regarding Circulatory Conditions among Swedish Patients in Opioid Substitution Treatment, with and without On-Site Primary Healthcare

**DOI:** 10.3390/ijerph18094614

**Published:** 2021-04-27

**Authors:** Eric Bäckström, Katja Troberg, Anders Håkansson, Disa Dahlman

**Affiliations:** 1Center for Primary Health Care Research, Department of Clinical Sciences, Lund University, Region Skåne, 205 02 Malmö, Sweden; ericbackstrom@gmail.com; 2Division of Psychiatry, Department of Clinical Sciences Lund, Lund University, 221 85 Lund, Sweden; katja.troberg@med.lu.se (K.T.); anders_c.hakansson@med.lu.se (A.H.); 3Malmö Addiction Centre, Skåne University Hospital, 205 02 Malmö, Sweden

**Keywords:** opiate substitution treatment, cardiovascular diseases, health services accessibility, Sweden, health equity, primary health care

## Abstract

Patients in Opioid Substitution Treatment (OST) have increased mortality and morbidity, with circulatory conditions suggested to be a contributing factor. Since OST patients tend to have unmet physical healthcare needs, a small-scale intervention providing on-site primary healthcare (PHC) in OST clinics was implemented in Malmö, Sweden in 2016. In this study, we assessed registered circulatory conditions and healthcare utilization in OST patients with and without use of on-site PHC. Patients from four OST clinics in Malmö, Sweden, were recruited to a survey study in 2017–2018. Medical records for the participants were retrieved for one year prior to study participation (*n* = 192), and examined for circulatory diagnoses, examinations and follow-ups. Patients with and without on-site PHC were compared through descriptive statistics and univariate analyses. Eighteen percent (*n* = 34) of the sample had 1≤ registered circulatory condition, and 6% (*n* = 12) attended any clinical physiology examination or follow-up, respectively. Among patients utilizing on-site PHC (*n* = 26), the numbers were 27% (*n* = 7) for circulatory diagnosis, 15% (*n* = 4) for examinations, and 12% (*n* = 3) for follow-up. OST patients seem underdiagnosed in regard to their circulatory health. On-site PHC might be a way to diagnose and treat circulatory conditions among OST patients, although further research is needed.

## 1. Introduction

People with opioid dependence have increased mortality compared to the general population, with contributing causes including overdosing, suicide and chronic diseases [1,2]. While opioid substitution treatment (OST) with buprenorphine or methadone effectively decreases mortality [3,4], both drug-related and non-drug-related deaths are over-represented among OST patients compared to the general population [5,6]. Suggested reasons for the increased mortality have been circulatory diseases in combination with potent substitution medication, especially among patients older than 55 years [5]. In a Scottish, longitudinal study, Gao et al. [5] conclude that “Circulatory disease is the co-morbidity most likely implicated in the quadrupling of methadone-specific DRD-risk [DRD = drug related death] at 45+ years; followed by digestive disease”.

While circulatory diseases are the primary causes of death in high-income countries, resulting in almost one third of all deaths in Sweden in 2016 [7], prevalence among patients in OST is sparsely investigated. Opioid use has been shown to be an independent risk factor for increased arterial stiffness, with a dose–response relationship [8]. One study associated consumption of opium via smoking and oral intake with increased severity of atherosclerosis, a risk factor for coronary artery disease [9]. There is also research showing that OST patients who receive pharmacological treatment with methadone are at an increased risk of having cardiovascular events through a QT interval prolongation [10,11]. People who inject drugs have a significantly increased risk for infective endocarditis [12], which may also have late effects after entering OST. In a review article, Robertson et al. describe a range of adverse vascular effects among people who inject drugs, including vascular infections, chronic venous insufficiency, and thrombosis/embolus [13].

Physical health in OST patients has been sparsely examined [14,15]. In a sample of Australian OST patients, Islam et al. [14] showed frequent electrocardiogram (ECG) abnormalities, chronic obstructive pulmonary disease (COPD) and liver disease. A recent Spanish study on patients receiving methadone substitution showed high numbers of metabolic syndrome; almost a third filled the criteria for metabolic syndrome and 37% had high blood pressure [16]. Exposure time to methadone therapy was associated with metabolic syndrome, with an increased odds ratio per 5 years of OST (odds ratio (OR): 1.38, 95% confidence interval (CI): 1.28–1.48) [16]. Tobacco smoking is an issue among a vast majority (approximately 80%) of OST patients [17,18,19,20], and is why chronic respiratory and circulatory diseases are likely to be over-represented. Mehta et al. [20] showed that methadone prescription was a predictor for both COPD and asthma, after adjusting for smoking status and smoking intensity, and suggested that heroin or crack smoking might be a confounder. Aging OST populations pose a challenge due to, e.g., increased medical comorbidities and cognitive impairment [15,21,22]. Research from the U.K. has shown that the risk of methadone-specific death among OST patients increases with age, in contrast to heroin-specific death [23].

In addition to OST patients having increased comorbidities compared to the general population, many of them lack a primary healthcare (PHC) contact. Previous research has shown that people who use illicit drugs are medically underserved and experience barriers towards healthcare seeking [24,25,26,27]. A recent study of OST patients in Malmö, Sweden, demonstrates that many worry about their somatic health and have had different health issues in the last year for which they have not contacted any healthcare services [18]. Factors contributing to unmet healthcare needs among OST patients might be the documented high rates of psychiatric comorbidity and psychosocial difficulties [28]. Older, international studies have shown positive results from linking addiction treatment with physical healthcare including PHC, regarding treatment of infectious and chronic diseases [29,30,31,32,33].

In 2014 a small-scale intervention providing adjusted, patient-centered visits at a primary healthcare center for patients in OST was introduced in Malmö, Sweden [34]. In 2016, patients at two OST clinics in Malmö were offered on-site appointments with a PHC physician from one designated PHC center. At the time of data collection for this study, on-site PHC at the OST clinic was offered at two OST clinics. The intervention consists of one PHC physician who schedules appointments every second week with patients on-site in the OST clinics, and also provides drop-in appointments. If the patient approves, there is close collaboration and information sharing between OST staff and the PHC physician. The intervention aims to minimize unmet healthcare needs regarding somatic issues among OST patients who, due to psychiatric or other obstacles, have difficulties contacting their PHC center and complying with booked appointments, medication and follow-up. It is voluntary for OST patients to utilize on-site PHC, or attend PHC as usual.

The aim of this study was to assess registered circulatory conditions in a sample of Swedish OST patients, and compare healthcare utilization for circulatory problems among OST patients with and without use of on-site PHC.

## 2. Materials and Methods

This was a retrospective medical record study on OST patients in Malmö, Sweden. The use of the journal data for this study has been approved by the Regional Ethics Board, Lund, Sweden, EPN file no. 2016/1105.

### 2.1. Setting

The study was conducted in Malmö, the third largest city in Sweden with more than 300,000 inhabitants, located in Skåne county (pop. 1.35 million) in southern Sweden.

By the time that the study was conducted, there were five OST clinics in Malmö providing treatment to approximately 520 opioid-dependent patients. In Sweden, OST clinics can be run by public as well as private caregivers, but all are tax-financed, covered by the Swedish universal health insurance, and provided at specialized psychiatric treatment units. Swedish OST clinics accept people who are a minimum of 20 years of age who are considered to have had an opioid dependence for at least one year. In addition to the pharmacological treatment with buprenorphine or methadone, OST includes psychological or psychosocial treatment, and regular testing for blood borne infections. OST staff include psychiatrists, nurses, and social workers. OST availability in Sweden differs depending on geographical region. Skåne county has 20 OST clinics, and thus high OST availability.

In Sweden, physical healthcare (primary, secondary and tertiary) is covered by the universal health insurance and tax-financed [35], which makes the healthcare strongly subsidized for the individual. There are both public and private healthcare providers. Circulatory conditions can be diagnosed and treated in specialized clinics (emergency clinic, hospital wards or specialized outpatient clinics) or in PHC. The latter is comprehensive, and all Swedish citizens are automatically registered at a PHC center (which can be actively changed by the individual).

### 2.2. Participants

Patients from four OST clinics in Malmö, Sweden, were recruited to a survey study in 2017–2018, as described by Troberg et al. [18]. By giving informed consent to study participation, the subjects also gave consent to a subsequent patient record analysis one year prior to inclusion in the study. The only inclusion criterion was receiving OST with methadone, buprenorphine or buprenorphine/naloxone, in Malmö. Patients who were unable to give informed consent due to drug influence or any other reason were excluded. There was no monetary compensation involved in the participation of the study. Two of the four OST clinics used for recruitment of study participants had access to on-site PHC at the time of survey collection.

### 2.3. Procedures 

Data regarding age, sex, country of birth, main source of income, housing status and smoking habits were collected through a self-report survey that was distributed at four OST clinics, as described by Troberg et al. [18].

Medical records from specialized clinics (E.D., inpatient and outpatient) and PHC in Skåne county for the 218 study participants were retrieved for one year prior to study participation for each individual. Since patients in OST per definition are registered in the specialized care records, all subjects for whom access to specialized care records was not allowed (due to confidentiality requested by the patient or invalid ID) were excluded from further analysis. Of the 218 study participants, 26 were excluded due to invalid ID (*n* = 25) or confidentiality requested by the patient (*n* = 1).

Patient records from specialized care were read digitally, while PHC records were retrieved as paper copies. Each record was searched for circulatory diagnoses, clinical physiology examinations and follow-ups due to circulatory problems, and utilization of on-site PHC in OST.

#### 2.3.1. Registrations for Circulatory Problems

All patient records from specialized and primary care were searched by the first author (E.B.) for the following ICD-10 diagnoses, noted as ICD-10 code or plain text:

G45 Transient cerebral ischemic attacks and related syndromes

G46 Vascular syndromes of brain in cerebrovascular diseases

I00-I02 Acute rheumatic fever

I05-I09 Chronic rheumatic heart diseases

I10-I16 Hypertensive diseases

I20-I25 Ischemic heart diseases

I26-I28 Pulmonary heart disease and diseases of pulmonary circulation

I30-I52 Other forms of heart disease

I60-I69 Cerebrovascular diseases

I70-I79 Diseases of arteries, arterioles and capillaries

R00 Abnormalities of heartbeat

R01 Cardiac murmurs and other cardiac sounds

R03 Abnormal blood-pressure reading, without diagnosis

R09 Other symptoms and signs involving the circulatory and respiratory system.

#### 2.3.2. Healthcare Utilization for Circulatory Problems

If any circulatory diagnosis was found, data were gathered on clinical physiology examinations and patterns for follow-ups including referrals sent to specialist clinics and PHC.

Clinical physiology examinations were defined as exercise ECG, echocardiography, long term ECG or myocardial perfusion imaging.

Patient attendance (yes or no) at any follow-up regarding their circulatory health was examined. The patients were viewed as having been to a follow-up if they either had a referral regarding their circulatory health to a PHC physician, cardiologist, neurologist or vascular surgeon and attended that appointment, or attended a planned follow-up without any referrals at the same clinic as they attended before.

#### 2.3.3. On-Site PHC Utilization

On-site PHC in OST was defined as any notes in the primary care records that the patient had had an appointment with a PHC physician at the OST clinic. These notes were exclusively written by the only physician providing on-site PHC in OST at the time of the study.

### 2.4. Analysis

Prior to analysis, three variables, with multiple-choice answers in the survey, were dichotomized: Housing situation was recoded to “unstable housing” if the respondent replied: “transitional apartment”, “institution/family care placement”, “hotel”, “homeless” or “other”. Main source of income was recoded to “unstable income” if the respondent replied: “public assistance” or “other”, while “employment”, “old age pension”, “sick leave” and “permanent sick leave” was recoded to “stable income”. Smoking habits was recoded to “current smoker” if the answer was “smoke daily” or “smoke less than daily”. Age was divided into three groups: <36 years, 36–50 years, and >50 years.

The collected data were organized in SPSS Statistics Version 26 ( IBM Corp. Released 2019. IBM SPSS Statistics for Windows, Version 26.0. Armonk, NY: IBM Corp) and analyzed through descriptive statistics. The data of the group of patients with access to on-site PHC were compared with the group of patients without on-site PHC through Chi-2 test, Fisher’s exact test, or univariate logistic regression (for categorical variables). *p* < 0.05 was considered statistically significant.

## 3. Results

### 3.1. Sample Characteristics

Of the 192 participants (71% male, median age 43 years (range 23–65; standard deviation 10.1 years)), a majority (74%) were born in Sweden (Table 1). Unstable housing was reported by 21%, unstable income by 59%, and tobacco smoking by 82%.

One hundred and thirty study participants (68%) had been in contact with PHC in Skåne county during the study period, and 26 (14%) had utilized on-site PHC in OST. Forty-two percent of the sample (*n* = 81) received OST at a clinic that had access to on-site PHC.

### 3.2. Healthcare Contacts Regarding Circulatory Conditions

One or more circulatory conditions were noted for 34 individuals (18%, Table 2). Hypertension (I10–I16) was diagnosed in 11 individuals (6%), of which six were older than 50 years, and an abnormal blood pressure reading was found in two. Six individuals had registered cardiovascular symptoms (palpitations, *n* = 4; unspecified tachycardia, *n* = 1; unspecified bradycardia, *n* = 1). Transient or chronic cerebrovascular diseases were diagnosed in five individuals (G45, *n* = 1; I60–I69, *n* = 4). All other circulatory diagnoses were noted in 0–3 individuals, respectively (angina pectoris *n* = 2, heart failure *n* = 2, tricuspid valve insufficiency *n* = 1, unspecified atrial fibrillation and atrial flutter *n* = 1, thoracic aortic aneurysm *n* = 1, aneurysm of an upper extremity artery *n* = 1).

A minority of the study sample (*n* = 12, 6%), underwent one or more of the clinical physiology examinations echocardiography (*n* = 8), exercise ECG (*n* = 2), long term ECG (*n* = 5) or myocardial perfusion imaging (*n* = 1) during the study period (Table 2).

In total, 12 individuals (6% of the total sample; 35% of those with a registered circulatory condition) had been to a follow-up regarding their circulatory condition, during the year investigated.

### 3.3. Correlates of On-Site PHC

The study subjects who utilized on-site PHC (*n* = 26) reported significantly more unstable housing than those not utilizing on-site care (OR 3.25; 95% CI 1.36–7.78; *p* = 0.006) (Table 3). There were no statistically significant correlations between on-site PHC utilization and age, sex, smoking, main income or country of birth.

Seven individuals (27%) among those with on-site PHC had registered circulatory conditions, compared with 16% among those with PHC as usual. A statistically significant difference of registered circulatory conditions between the groups was not found (OR 1.90; 95% CI 0.73–4.95; *p* = 0.27).

Four individuals utilizing on-site PHC (15%) attended an examination for circulatory conditions, compared to eight individuals with PHC as usual (5%), (OR 3.59; 95% CI 1.00–12.92; *p* = 0.06). Circulatory follow-ups were registered among 12% (*n* = 3) in the on-site group and 5% (*n* = 9) in the regular PHC group (OR 2.28; 95% CI 0.57–9.03; *p* = 0.21).

## 4. Discussion

This study shows surprisingly low rates of circulatory diagnoses (18%) among patients in OST. Even though the small sample size in this study limits the conclusions that can be drawn, the results imply that more research is needed. Even a common condition like hypertension was only noted in six percent of the study sample and 11% among those older than 50 years. These numbers are drastically lower than anticipated, since previous research has found the prevalence of hypertension to be 38% in adults in Sweden [36], or 27% of the Swedish general population aged 20 years or older [37].

It is implausible that our findings reflect the true prevalence of circulatory diseases in the OST group, given the high percentage of tobacco smoking (82%; 75% daily smoking) compared to 7% daily smoking in the general Swedish population [38]. Other lifestyle factors known to increase the risk of circulatory disease are common among patients in OST according to previous research, such as overweight [39,40] and prior or current use of other substances such as alcohol [41] and stimulants [42]. Sweeney et al. showed that the prevalence of diabetes, hypercholesterolemia and hypertension increased with time in methadone treatment [40], and Vallecillo et al. found that 37% of patients receiving methadone substitution had high blood pressure when measuring [16].

Thus, we suggest under-diagnosing of circulatory conditions in the OST patient group, which could be suggested when interpreting some studies analyzing healthcare seeking patterns and barriers to the use of healthcare among people with SUD [18,24,25,26,27]. Further, we do not suspect that poor documentation explains the low diagnosis rates, since diagnoses in the national Swedish inpatient register have been shown to be valid in 85–95% of cases [43]. Another plausible reason for the relatively low prevalence is that the OST patients often seek medical attention for symptoms which, for example, hypertension does not really have. Additionally, hypertension is a condition often detected by occupational health services, which does not apply for a majority of OST patients, due to unemployment.

Our finding that clinical physiology examinations and follow-ups seem to be uncommon in the sample, is in line with data from Spithoff et al. showing that patients in OST were less likely to receive diabetes monitoring than matched controls [44]. In our record data, it was not possible to see if the patients were referred and booked for examinations and follow-up and missed their appointments, of if no intervention was planned.

In this study, the percentage of individuals with registered circulatory conditions in the on-site PHC group (27%) was closer to that of the general Swedish population. The percentage of patients attending examinations and follow-up was higher in the on-site group compared to the group utilizing PHC as usual. Although there were only 26 patients in the on-site group, and no statistically significant correlations were found, we hypothesize that on-site PHC could be a way to diagnose more circulatory diseases in the OST patient group. Future research is needed to evaluate effects on diagnosis rates and patient health, from on-site PHC in OST. We found that those who utilized on-site PHC had significantly more unstable housing than those with PHC as usual, which indicates socioeconomic vulnerability in the target group. Improving somatic health among patients with OST is connected to an array of challenges on many levels. The marginalization and stigma attached to the OST population [45], combined with a high degree of psychiatric comorbidity [28] and polysubstance use [41,42], often in combination with criminal behavior in order to finance other dependencies other than opioids, all add to the difficulties of prioritizing somatic health from the individual standpoint.

This study has limitations. The number of study participants was small, especially in the group with on-site PHC, which makes it difficult to draw certain conclusions from the results and statistical analyses. A larger study sample, or a comparison with a control group in a different clinical setting, would allow more refined analyses including subgroup analysis by the generic OST drug (methadone, buprenorphine, buprenorphine/naloxone). Since the patient charts were reviewed for no more than one year, we might have excluded some cases with diagnosed circulatory conditions but less than yearly healthcare visits. In addition, the results may not be fully generalizable outside Skåne county. However, the sample is considered fairly representative for the OST population in Malmö. Our study sample made up 37% of the total number of OST patients in Malmö by the time of study inclusion, and differences between our sample and the total OST population regarding gender distribution (male sex: 71% vs. 72%) and median age (43 years vs. 45 years) were small. Swedish healthcare is comprehensive and tax financed [35] and Skåne county has high OST availability. Still, the OST patient population in Malmö has a high burden of psychiatric and physical morbidity, comparable to international data. Previous research from Skåne county has shown that 72% of OST patients in Malmö had experienced at least one overdose and 31% had tried to commit suicide [46].

Even though this was a small study, our results have important implications. Our findings imply a need for larger, longitudinal studies on OST patients’ cardiovascular health and healthcare contacts. As OST patients appear to be underdiagnosed concerning their circulatory health, it might be a good idea to establish routines addressing that issue. The OST patients often visit psychiatric or addiction clinics but do not often receive physical examinations. If the psychiatric, addiction and somatic caregivers were more integrated, more of the somatic diseases would probably be found in OST patients and they would probably also be found in earlier stages. One method of doing this could be measuring blood pressure at enrollment to the OST. Offering yearly basic physical examinations for OST patients, possibly combined with on-site PHC, might benefit the continuity of care for the OST patients. It is noteworthy that the strategy of combining drug treatment and primary care is stressed in, e.g., the Swedish and U.K. national guidelines on drug misuse and dependence [47,48]. We also suggest that on-site PHC needs to be evaluated on a larger scale. Although our results are tentative and based on a small number of individuals, the data indicate that its use leads to more frequent diagnosis and better compliance to examinations and follow-up.

## 5. Conclusions

This study was one of the first to assess healthcare contacts for circulatory conditions among patients in OST, and showed that OST patients seem underdiagnosed in regard to their circulatory health. On-site PHC in OST attracts socioeconomically challenged patients, and might be useful for diagnosing and treating circulatory conditions among people with opioid dependence.

## Figures and Tables

**Table 1 ijerph-18-04614-t001:** Sample characteristics. *n* = 192.

Characteristic	*n* (%)	Years (Range; Standard Deviation)
Median age		43 (23–65; 10.1)
Age group		
23–35 years	40 (21)	
36–50 years	99 (52)	
51–65 years	53 (28)	
Male sex	136 (71)	
Country of birth		
Sweden	142 (74)	
Other country	48 (25)	
Missing	2 (1)	
Housing situation		
Unstable	41 (21)	
Stable	148 (77)	
Missing	3 (2)	
Main source of income		
Unstable	113 (59)	
Stable	73 (38)	
Missing	6 (3)	
Smoking habits		
Current smoker	157 (82)	
Not current smoker	28 (15)	
Missing	7 (4)	
Primary care record entry	130 (68)	
OST clinic with on-site PHC available	81 (42)	
Access to on-site PHC	26 (14)	

OST = opioid substitution treatment. PHC = primary healthcare.

**Table 2 ijerph-18-04614-t002:** Registered diagnoses and healthcare utilization for circulatory conditions.

Diagnosis	Diagnosis *n* (%)	Clinical Physiology Examination *n* (%)	Follow-Up *n* (%)
Any circulatory diagnosis	34 (18)	12 (6)	12 (6)
G45 Transient cerebral ischemic attacks and related syndromes	1 (<1)	1 ^a^	1
G46 Vascular syndromes of brain in cerebrovascular diseases	0	N/A	N/A
I00–I02 Acute rheumatic fever	0	N/A	N/A
I05–I09 Chronic rheumatic heart diseases	1 (<1)	1 ^b^	1
I10–I16 Hypertensive diseases	11 (6)	2 ^b^	3
I20–I25 Ischemic heart diseases	2 (1)	0	1
I26–I28 Pulmonary heart disease and diseases of pulmonary circulation	2 (1)	0	1
I30–I52 Other forms of heart disease	3 (2)	1 ^b^	1
I60–I69 Cerebrovascular diseases	4 (2)	1 ^a^	1
I70–I79 Diseases of arteries, arterioles and capillaries	2 (1)	1 ^b^	2
R00 Abnormalities of heart beat	6 (3)	4 ^c^	1
R01 Cardiac murmurs and other cardiac sounds	0	N/A	N/A
R03 Abnormal blood-pressure reading, without diagnosis	2 (1)	1 ^d^	0
R09 Other symptoms and signs involving the circulatory and respiratory system	0	N/A	N/A

^a^ Echocardiography + long term ECG. ^b^ Echocardiography. ^c^ Long term ECG (*n* = 3); Echocardiography + exercise ECG (*n* = 1). ^d^ Myocardial perfusion imaging. *n* = 192. ECG = electrocardiography. N/A = not applicable.

**Table 3 ijerph-18-04614-t003:** Comparison of patients with and without access to on-site primary health care.

Characteristic	PHC as Usual *n* (%)	On-Site PHC *n* (%)	OR (95% CI)	*p*-Value
*n*	166 (100)	26 (100)		
Age group ^a^				
<36 years (reference)	33 (20)	7 (27)	1.00	N/A
36–50 years	90 (54)	9 (35)	0.47 (0.16–1.37)	0.17
>50 years	43 (26)	10 (39)	1.10 (0.38–3.19)	0.87
Male sex	120 (72)	16 (62)	0.61 (0.26–1.45)	0.27
Born in Sweden	126 (76) ^1^	16 (62)	2.07 (0.87–4.94)	0.10
Unstable housing	30 (18) ^2^	11 (42)	3.25 (1.36–7.78)	0.006
Unstable income	98 (59) ^3^	15 (58) ^4^	1.09 (0.45–2.64)	0.85
Current tobacco smoking	135 (81) ^5^	22 (85)	0.98 (0.31–3.09)	0.97
Circulatory diagnosis ^b^	27 (16)	7 (27)	1.90 (0.73–4.95)	0.27
Clinical physiology examination ^b^	8 (5)	4 (15)	3.59 (1.00–12.92)	0.06
Cardiovascular follow-up ^b^	9 (5)	3 (12)	2.28 (0.57–9.03)	0.21

^a^ Univariable logistic regression analysis. ^b^ Fisher’s exact test. ^1^ missing *n* = 2. ^2^ missing *n* = 3. ^3^ missing *n* = 4. ^4^ missing *n* = 2. ^5^ missing *n* = 7. Chi-square test. *n* = 192. OR = Odds Ratio. CI = Confidence Interval. N/A = not applicable. PHC = primary healthcare.

## Data Availability

The SPSS data used to support the findings of this study are restricted by the Regional Ethics Board, Lund, Sweden, in order to protect patient privacy. Data are available from Disa Dahlman, disa.dahlman@med.lu.se, for researchers who meet the criteria for access to confidential data.

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
