# Peer review of "Healthcare Contacts Regarding Circulatory Conditions among Swedish Patients in Opioid Substitution Treatment, with and without On-Site Primary Healthcare"

_ijerph, 2021, doi:10.3390/ijerph18094614_

Round 1
Reviewer 1 Report
- First of all, the author lacks a strong validation of why they conducted the study. From the key supporting literature [5], the original take-home message from this literature was “Circulatory disease is the co-morbidity most likely implicated in the quadrupling of methadone-specific DRD-risk at 45+ years; followed by digestive disease”. But the authors wrote, “Neoplasms, respiratory and digestive diseases are also suggested to affect the mortality rates [5]”.
The literature [11] found “methadone may be associated with QTc prolongation, whereas buprenorphine may not.” and the literature [10] confirmed “Methadone is associated with prolonged QT intervals, but there was no association with dose. Buprenorphine did not prolong the QT interval.” If this is the case, the researchers should be cautious in defining “opioid substitution treatment (OST)” patient, in terms of the type of specific drug (buprenorphine, methadone, and naltrexone). A subgroup analysis by the generic drug (buprenorphine, methadone, and naltrexone) is strongly recommended. - The other limitation of this research is the small sample size, and therefore the small counts for diagnoses for circulatory conditions. This study aimed to assess registered circulatory conditions in a sample of Swedish OST patients, and compare healthcare utilization for circulatory problems 93 among OST patients with and without the use of on-site PHC. Probably due to the small number of circulatory problems, the authors could not carry on with more advanced statistical analysis, for example, the Cox regression to generate the hazard ratios of incidence of circulatory conditions. The authors concluded “This study shows surprisingly low rates of circulatory diagnoses (18%) among patients in OST. Even a common condition like hypertension was only noted in six percent 256 of the study sample and 11% among those older than 50 years.” It bears mentioning, the “lower rates” has very minor weight in delivering major conclusions and recommendations on the issue of “circulatory conditions in OST patients”.
- The measurement of “On-site PHC in OST patients” as any notes for the appointment with a PHC physician at the "OST clinic" was inaccurate. And the conclusion “clinical physiology examinations and follow-ups were uncommon in the sample” from this measurement is barely convincing.
Overall, there must be a lot of improvement in the measurement, statistical analysis, and academic writing to avoid biased clinical implications.
Author Response
Reviewer 1
1.First of all, the author lacks a strong validation of why they conducted the study. From the key supporting literature [5], the original take-home message from this literature was “Circulatory disease is the co-morbidity most likely implicated in the quadrupling of methadone-specific DRD-risk at 45+ years; followed by digestive disease”. But the authors wrote, “Neoplasms, respiratory and digestive diseases are also suggested to affect the mortality rates [5]”.
The literature [11] found “methadone may be associated with QTc prolongation, whereas buprenorphine may not.” and the literature [10] confirmed “Methadone is associated with prolonged QT intervals, but there was no association with dose. Buprenorphine did not prolong the QT interval.” If this is the case, the researchers should be cautious in defining “opioid substitution treatment (OST)” patient, in terms of the type of specific drug (buprenorphine, methadone, and naltrexone). A subgroup analysis by the generic drug (buprenorphine, methadone, and naltrexone) is strongly recommended.
Response: Thank you for valuable feedback on our manuscript! We have adjusted the paper according to the suggestions given, and feel that it has improved.
Regarding the citation above (“Neoplasms, respiratory and digestive diseases are also suggested to affect the mortality rates [5]”.), our intention was to mention more than one suggested reason for the increased mortality rates among patients in OST. However, we agree that the rationale behind the study was somewhat vague. We appreciate the reviewer’s critique and have revised the Introduction in order to stress the importance of cardiovascular disease in this patient group. On p. 1, we have now written ”Suggested reasons for the increased mortality have been circulatory diseases in combination with potent substitution medication, especially among patients older than 55 years [5]. In a Scottish, longitudinal study, Gao et al. [5] conclude that ‘Circulatory disease is the co-morbidity most likely implicated in the quadrupling of methadone-specific DRD-risk [DRD=drug-related death] at 45+ years; followed by digestive disease.’”
Regarding OST medications’ effect on QTc prolongation, we have clarified (p. 2) that ”There is also research showing that OST patients who receive pharmacological treatment with methadone are at an increased risk of having cardiovascular events through a QT interval prolongation [10,11].“
If we had had a larger study sample, a subgroup analysis by generic drug would have been of interest. We have stressed this limitation in the Discussion (p. 8-9): ”This study has limitations. The number of study participants was small, especially in the group with on-site PHC, which makes it difficult to draw certain conclusions from the results and statistical analyses. A larger study sample, or a comparison with a control group in a different clinical setting, had allowed more refined analyses including subgroup analysis by the generic OST drug (methadone, buprenorphine, buprenorphine/naloxone).”
2.The other limitation of this research is the small sample size, and therefore the small counts for diagnoses for circulatory conditions. This study aimed to assess registered circulatory conditions in a sample of Swedish OST patients, and compare healthcare utilization for circulatory problems 93 among OST patients with and without the use of on-site PHC. Probably due to the small number of circulatory problems, the authors could not carry on with more advanced statistical analysis, for example, the Cox regression to generate the hazard ratios of incidence of circulatory conditions. The authors concluded “This study shows surprisingly low rates of circulatory diagnoses (18%) among patients in OST. Even a common condition like hypertension was only noted in six percent 256 of the study sample and 11% among those older than 50 years.” It bears mentioning, the “lower rates” has very minor weight in delivering major conclusions and recommendations on the issue of “circulatory conditions in OST patients”.
Response: Please see response to comment 1 above. We have also added in the Discussion (p. 9) ”Even though this was a small study, our results has important implications. Our findings imply a need for larger, longitudinal studies on OST patients’ cardiovascular health and healthcare contacts.”, and commented the small sample size on p. 8: “Even though the small sample size in this study limit the conclusions that can be drawn, the results imply that more research is needed.”
3.The measurement of “On-site PHC in OST patients” as any notes for the appointment with a PHC physician at the "OST clinic" was inaccurate. And the conclusion “clinical physiology examinations and follow-ups were uncommon in the sample” from this measurement is barely convincing.
Response: We have tried to clarify the definition of on-site PHC by stating (p. 4) ”These notes were exclusively written by the only physician providing on-site PHC in OST by the time of the study.” If we have misinterpreted the reviewer’s comment, we are happy to revise this section further.
Given the small study sample, we hade adjusted the sentence above (p. 8): ”Our finding that clinical physiology examinations and follow-ups seem to be uncommon in the sample…”
Overall, there must be a lot of improvement in the measurement, statistical analysis, and academic writing to avoid biased clinical implications.
Response: We agree with the reviewer that this is a small study, where there is limited support for clear conclusions and clinical implications. Since OST patients’ physical health in general and cardiovascular health/healthcare contacts in particular are under-researched areas, we feel that also a small clinical study such as ours is of interest as an indicator of future research needs. We have tried to stress the study’s limitations – but also rationale and value – in accordance with the reviewer’s comments (please see responses above).
Reviewer 2 Report
This is an interesting paper in an area of clinical practice which, as the authors say, is under investigated. The problem is that it is a small study sample and there are, obviously, no control groups. this is always a problem for clinical observations of this sort and inevitably limit the conclusions that can be drawn. the authors do say that more research is needed and I would draw attention to our recent publication on vascular disease in injecting drug users, but they need to be careful in saying that no one has observed vascular problems in primary care (citation below). The authors are correct in saying more research is needed and I am also pleased to hear them say that Primary care is an essential element of intervention for this group. The UK national guidelines clearly say that shared care and primary care is essential (citation below).
Robertson R, Broers B, Harris M. Injecting drug use, the skin and vasculature. Addiction. 2020 Oct 13. doi: 10.1111/add.15283. Epub ahead of print. PMID: 33051902.
Clinical Guidelines on Drug Misuse and Dependence Update 2017 Independent Expert Working Group (2017) Drug misuse and dependence: UK guidelines on clinical management. London: Department of Health
Author Response
Reviewer 2
This is an interesting paper in an area of clinical practice which, as the authors say, is under investigated. The problem is that it is a small study sample and there are, obviously, no control groups. this is always a problem for clinical observations of this sort and inevitably limit the conclusions that can be drawn. the authors do say that more research is needed and I would draw attention to our recent publication on vascular disease in injecting drug users, but they need to be careful in saying that no one has observed vascular problems in primary care (citation below). The authors are correct in saying more research is needed and I am also pleased to hear them say that Primary care is an essential element of intervention for this group. The UK national guidelines clearly say that shared care and primary care is essential (citation below).
Robertson R, Broers B, Harris M. Injecting drug use, the skin and vasculature. Addiction. 2020 Oct 13. doi: 10.1111/add.15283. Epub ahead of print. PMID: 33051902.
Clinical Guidelines on Drug Misuse and Dependence Update 2017 Independent Expert Working Group (2017) Drug misuse and dependence: UK guidelines on clinical management. London: Department of Health
Response: We are grateful for the reviewer’s positive critique, and appreciate the suggestions given. We have adjusted the paper accordingly, and feel that it has improved.
Firstly, we have stressed the limitations of the study (small sample size etc.) more explicitly in the Discussion, p. 8-9: “This study has limitations. The number of study participants was small, especially in the group with on-site PHC, which makes it difficult to draw certain conclusions from the results and statistical analyses. A larger study sample, or a comparison with a control group in a different clinical setting, had allowed more refined analyses including subgroup analysis by the generic OST drug (methadone, buprenorphine, buprenorphine/naloxone).” and “”Even though this was a small study, our results has important implications. Our findings imply a need for larger, longitudinal studies on OST patients’ cardiovascular health and healthcare contacts…”. We have also tried to state the main results in a slightly more careful manner. For example, we have reworded the conclusion “clinical physiology examinations and follow-ups were uncommon in the sample” as “clinical physiology examinations and follow-ups seem to be uncommon in the sample” (p. 8), and added the sentence “Even though the small sample size in this study limit the conclusions that can be drawn, the results imply that more research is needed” on p. 8. These changes was also according to the comments from reviewer 1.
We appreciate the citations provided by the reviewer. We have included the study by Robertson et al in the Introduction (p. 2): “In a review article, Robertson et al. describe a range of adverse vascular effects among people who inject drugs, including vascular infections, chronic venous insufficiency, and thrombosis/embolus [13].”. The UK national guidelines are now mentioned in the Discussion (p. 9): “It is noteworthy that the strategy of combining drug treatment and primary care is stressed in, e.g., the Swedish and U.K. national guidelines on drug misuse and dependence.”.
Reviewer 3 Report
Authors assessed registered circulatory conditions and healthcare utilization in opioid substitution treatment (OST) patients with and without use of on-site primary healthcare (PHC). And their data shows that OST patients seem underdiagnosed in regard to their circulatory health and on-site PHC might be a way to diagnose and treat circulatory conditions among OST patients. Their data should be interesting and helpful for safe of OST in people with opioid dependence. Although their article should be published for International Journal of Environmental Research and Public Health, authors must revise Discussion, as following,
Data were collected at only a provincial city in Sweden. Therefore, there may be differences in registered circulatory conditions and healthcare utilization in OST patients among area. And there may be differences in population of opioid dependence and medical system among country. Authors must describe about these information in other area or countries and compared these in Discussion. If authors can show data using references and discussions among area or country, subscribers will be interested in their data and conclusion. This journal is international, so that there are many subscribers in the world.
Author Response
Reviewer 3
Authors assessed registered circulatory conditions and healthcare utilization in opioid substitution treatment (OST) patients with and without use of on-site primary healthcare (PHC). And their data shows that OST patients seem underdiagnosed in regard to their circulatory health and on-site PHC might be a way to diagnose and treat circulatory conditions among OST patients. Their data should be interesting and helpful for safe of OST in people with opioid dependence. Although their article should be published for International Journal of Environmental Research and Public Health, authors must revise Discussion, as following,
Data were collected at only a provincial city in Sweden. Therefore, there may be differences in registered circulatory conditions and healthcare utilization in OST patients among area. And there may be differences in population of opioid dependence and medical system among country. Authors must describe about these information in other area or countries and compared these in Discussion. If authors can show data using references and discussions among area or country, subscribers will be interested in their data and conclusion. This journal is international, so that there are many subscribers in the world.
Response: We thank the reviewer for valuable feedback. We have added the following section in the Discussion (p. 9): “The Swedish healthcare is comprehensive and tax financed [34] and Skåne county has high OST availability. Still, the OST patient population in Malmö has a high burden of psychiatric and physical morbidity, comparable to international data. Previous research from Skåne county has shown that 72% of OST patients in Malmö had experienced at least one overdose and 31% had tried to commit suicide [46].”